# Evaluation of the Absorption Behavior of Main Component Compounds of Salt-Fried Herb Ingredients in Qing’e Pills by Using Caco-2 Cell Model

**DOI:** 10.3390/molecules23123321

**Published:** 2018-12-14

**Authors:** Jinlan Lu, Ling Liu, Xingyu Zhu, Li Wu, Zhipeng Chen, Zisheng Xu, Weidong Li

**Affiliations:** 1College of Pharmacy, Nanjing University of Chinese Medicine, Nanjing 210046, China; 20161357@njucm.edu.cn (J.L.); ll6611412@126.com (L.L.); 15295512919@163.com (X.Z.); wuli87107@163.com (L.W.); czpcpu2000@hotmail.com (Z.C.); 2Engineering Center of State Ministry of Education for Standardization of Chinese Medicine Processing, Nanjing University of Chinese Medicine, Nanjing 210046, China; 3Wuhu Pure Sunshine Natural Medicine Company Limited, Wuhu 241000, China

**Keywords:** Qing’e Pills, Caco-2 cell, absorption, salt-fried process

## Abstract

Qing’e Pills is a Chinese traditional herbal product, which is often used to strengthen muscles and bones in TCM (traditional Chinese Medicine) practice. Its two main component herbs, namely, *Cortex Eucommiae* and *Fructus Psoraleae* are both required to be salt-fried according to TCM theory. We have evaluated the effects of salt-frying treated herbs on Caco-2 cell uptake behavior for those active ingredients of Qing’e Pills. By investigating of various variables, including MTT, temperature, inhibitors, pH, salt concentration and herb processing methods, we tried to clarify whether the salt-processing on herbs was necessary or not. Results showed that, compared to other processing methods, the salt-frying process significantly (*p* < 0.01) enhanced the absorption of effective components of Qing’e Pills. The way that psoralen, isopsoralen, psoralenoside and geniposide acid entered Caco-2 cells at low concentrations was via passive diffusion. These components were not substrates of P-glycoprotein. It demonstrated that the salt-frying process not only enhanced the concentration of active components in herb extract, but also changed their absorption behaviors. Nevertheless, the mechanism of absorption behavior changing needs to be further investigated.

## 1. Introduction

Qing’e Pills was first recorded in the Song Dynasty “Taiping Huimin and Agent Bureau” [1] of ancient Chinese Song Dynasty. It was collected in Chinese Pharmacopoeia (2015 Edition) [2]. At present, it is mainly used to treat osteoporosis. Its prescription composed of four herbs, i.e., *Cortex Eucommiae* (salt-fried), *Fructus Psoraleae* (salt-fried), *Semen Juglandis* (*walnut*, fried) and *Bulbus Allii Sativi* (*garlic*). 

Geniposide acid (GPA) can promote the differentiation and maturation of osteoblasts, and inhibit their transformation into osteoclasts. It is an active ingredient in *Cortex Eucommiae* for treating osteoporosis [3]. Psoralen (P) and isopsoralen (IP) are the active components of *Fructus Psoraleae*, which was typically used for osteoporosis treatment. Functioning as phytoestrogen [4], P can facilitate the differentiation and maturation of osteoblasts [5,6], and suppress their activity [7]. IP also has estrogenic effects, which selectively acts on estrogen receptor [8], promotes osteoblast differentiation and maturation [9], and synergizes with zinc to induce the expressions of transcription factors related to osteoblasts [10]. The effect-time and concentration-time curves of GPA and P exhibited high correlations, while the correlations of salt-processed Qing’e Pills exceed that of the raw one [11].

It was reported that by comparing to Qing’e Pills with raw herb ingredients, the Pills with salt-processed herb ingredients exhibited better therapeutic effect on osteoporosis for ovariectomized rats [12], though the content of GPA, P and IP did not increase significantly in the Pills with salt-processed herb materials [13]. This enhancement of efficacy may be due to the salt-processing affecting the absorption behavior of the main components of Qing’e Pills. It is reported that psoralenoside (PO) and isopsoralenoside (IPO) can be converted to P and IP respectively in the intestine [14,15]. In this research, the absorption behavior of these components was investigated using a Caco-2 cell model.

## 2. Results

### 2.1. Optimization of Experimental Conditions

To reach a high response, the electrospray ionization (ESI) source allowing simultaneous scan in the positive and negative ion modes was employed, and the multiple reaction monitoring (MRM) mode was used for quantitation. Detection of the analytes was optimized in both positive and negative ion modes. The detection results of P, IP, PO and escoparone (internal standard, IS) and those of GPA and rhein (IS) were much more acceptable in the positive ion mode and the negative ion mode respectively. In addition, mass parameters, i.e., declustering potential (DP) and collision energy (CE), were optimized to enhance the ionization efficiency. The DP values of P, IP, PO, escoparone (IS), GPA and rhein (IS) were 82.75, 91.05, 87.00, 93.08, −126.4 and −59.15 V respectively, and their CE values were 33.1, 30.54, 30.73, 27.23, −18.65 and −19.01 V respectively.

Chromatographic conditions were optimized to achieve high resolutions and sensitive signals of the analytes. After several mobile phase systems were tested, the acetonitrile/water (0.1% formic acid) system with ideal separation and peak symmetry was finally selected. 

To augment the reliability and reproducibility, it is necessary to select an optimum sample preparation method with high recovery and suitable matrix effect for all analytes. Since proteins can be easily precipitated, this method is suitable for detecting these components.

### 2.2. Method Validation

#### 2.2.1. Specificity

The chromatograms of blank cell lysate, cell lysate sample spiked with analyte and IS, and the real sample collected from cells after drug administration are shown in Figure 1. The retention time of the analyte, or IS, was hardly subjected to interference.

#### 2.2.2. Linearity and Lower Limit of Quantification

The calibration curve of each component was plotted by assaying the calibration samples at seven concentration levels. Typical equations for the calibration curves and the correlation coefficients (r) were y = 0.00786x + 0.00385 (r = 0.99984) for P, y = 0.01367x + 0.02159 (r = 0.99983) for IP, y = 0.00161x − 0.00245 (r = 0.99920) for PO, and y = 0.000131128x + 0.000119716 (r = 0.99958) for GPA. The lower limits of quantification for P, IP, PO and GPA in cell lysate were 5 ng/mL.

#### 2.2.3. Precision and Accuracy

The accuracy and precision were determined by using QC samples at low, middle and high concentrations respectively on three consecutive days. To study the precision, six duplicates were prepared. The intra-day and inter-day precisions were within 3.47–9.87% and 4.55–9.60%, respectively. The accuracy ranged between ~7.06% and ~0.05% for all analytes (Table 1). Clearly, all values were within the acceptable range, which means that the developed method was reliable, accurate and reproducible.

#### 2.2.4. Extraction Recovery and Matrix Effect

As shown in Table 2, the extraction recoveries of analytes and IS in cell lysate at three concentration levels all exceeded 72.64%, and the (relative standard deviation) RSD values were less than 15%. Besides, the matrix effect of all analytes and IS were within the acceptable range of 70.97–104.03%, and the RSD values were less than 15%. The results demonstrated that no significant interference in cell lysate existed.

#### 2.2.5. Stability

The stability was tested under three different conditions: three freeze-thaw cycles (−20 °C to room temperature), short-term stability (at room temperature for 4 h) and long-term stability (at −20 °C for 20 days) (Table 3). The obtained RSD values were all less than 11.31%. Therefore, there was no obvious substance loss throughout the period of experiment.

### 2.3. Cell Viability Test

The effects of drug-contained medium with different concentration of extracts of various Qing’e Pills on Caco-2 cell viability activity were presented in Figure 2. In the range of 5–20 mg/mL, the extract-contained medium promoted cell proliferation. However, when the concentration reached 80 mg/mL, the medium significantly suppressed the proliferation, with the inhibition rate of about 30%. Thus, low or over-high concentrations were not suitable for Caco-2 cell uptake experiment. The concentration of drug-contained medium at 40 mg/mL, it showed mildest effect on cell growth. Thus, this concentration was chosen for Caco-2 cellular uptake experiments.

### 2.4. Uptake by Caco-2 Cells

#### 2.4.1. Effects of Culture Media Containing Extracts with Different Concentrations on Cellular Uptake 

The effect of extract concentration on cellular uptake was evaluated by 60 min of incubation of Caco-2 cells with 500 μL of drug-contain medium of the extract of salt-fried Qing’e Pills at different concentrations (5, 40, 80, 120, 240 mg/mL; three parallel wells for each concentration). As shown in Figure 3, with the rising concentration from 5 to 240 mg/mL, the uptake of four compounds increases linearly, suggesting that P, IP, PO and GPA underwent uptake through passive diffusion in the range of 5–240 mg/mL.

#### 2.4.2. Effect of Temperature on Cellular Uptake

The effect of temperature on cellular uptake was investigated by 60 min of incubation Caco-2 cells with 500 μL of culture drug-contained medium of the extract of salt-fried Qing’e Pills at 4 °C and 37 °C (three parallel wells for each temperature). As exhibited in Figure 4, temperature hardly affects the uptake of the four compounds.

#### 2.4.3. Effect of Inhibitor on Cellular Uptake

The effect of inhibitor on cellular uptake was investigated by 60 min of incubation Caco-2 cells with 500 μL of drug-contain medium of the extract of salt-fried Qing’e Pills and inhibitors (concentrations of verapamil, cyclosporine A and sodium azide: 100, 10 and 500 μM respectively; three parallel wells for each inhibitor) at 37 °C. As shown in Figure 5, the inhibitors exhibited no significant effects on cellular uptake of the four compounds. 

Verapamil and cyclosporine A are P-glycoprotein inhibitors [16,17], and sodium azide is a strong metabolic inhibitor of the respiratory chain [18]. With the participation of the three inhibitors, the uptake of these components did not significantly increase or decrease. Thus, P, IP, PO and GPA were not the substrates of P-glycoproteins, without needing energy to enter cells. The results further verified that they entered Caco-2 cells through passive diffusion at the tested concentrations.

#### 2.4.4. Effect of pH on Cellular Uptake

The pH of the drug-contain medium of the extract of salt-fried Qing’e Pills was adjusted to 5.0, 6.0, 7.0 and 8.0 by adding 1 M HCl or NaOH solution. Subsequently, cells were incubated with 500 μL of the medium for 60 min at 37 °C (three parallel wells for each pH). Figure 6 exhibited that the uptake of P, IP and PO at pH 7.0 and 8.0 as well as that of GPA at pH 7.0 were higher than the uptake of the blank group.

The pH values of the human intestinal tract range between 6.0 and 8.0, and most of the intestinal segments are neutral or weakly alkaline [19]. Obviously, the intestinal environment is beneficial for the absorption of these compounds.

#### 2.4.5. Effect of Salt Concentration on Cellular Uptake

The effect of salt concentration on cellular uptake was investigated by 60 min of incubation of Caco-2 cells with 500 μL of drug-contain medium of the extract of salt-fried Qing’e Pills with different salt concentrations (0.2, 0.5, 1.0, 2.0 and 5.0%; three parallel wells for each concentration) at 37 °C. As shown in Figure 7, salt concentration at 5.0%, it barely affected the uptake of the four component compounds; while when salt concentration at 0.2–2.0%, the uptake of P and PO significantly increased. In the presence of 0.2% and 2.0% salts, the uptake of IP significantly increased too. When the salt concentration was 1.0%, the uptake of GPA was significantly higher than that of blank group. 

The salt concentration of Qing’e Pills is about 1.45%. When salt concentration between 1.0–2.0%, all of the 4 component compounds exhibited higher cellular uptake compared with control group. Hence, the salt amount is reasonable for this processing technology, and the salt can promote the absorption of the main component compounds of Qing’e Pills.

#### 2.4.6. Effect of Herb Processing Methods on Cellular Uptake

The effects of different herb processing methods on cellular uptake were evaluated by 60 min incubation of Caco-2 cells with 500 μL of drug-contained medium of the extracts that obtained from raw, fried, salt-soaked and salt-fried Qing’e Pills (three parallel wells for each method) at 37 °C. The uptake of IP and PO in fried extract was significantly lower than that of raw one (Figure 8). Moreover, the uptake of P, IP and GPA significantly increased after salt-soaking. Both of the absorption of PO and GPA was significantly enhanced in salt-frying group, as compared to raw. 

Results showed that the process of frying was not beneficial to the absorption of IP and PO; while salt-soaking improved the absorption of P, IP and GPA. This manifested that the traditional Chinese herb processing technique indeed affects the absorption of component compounds of herbs.

## 3. Discussion

The structural and biochemical functions of Caco-2 cell (the human colon adenocarcinoma cell lines) are similar to that of the human intestinal epithelial cell, with enzyme system similar to those of the brush-edge epithelium of the small intestine. Immortalized Caco-2 cell line is commonly used due to its ability to form an adherent monolayer, which exhibits some characteristics of the gut epithelium [20]. Caco-2 cells are widely used as a standard in vitro model for studying the absorption and metabolism by intestinal epithelial cells [21,22,23,24]. Specifically, the absorption of a compound by Caco-2 cells can reflect its absorption by the small intestine to a certain extent.

By studying the uptake behavior of P, IP, PO and GPA, we found that they entered Caco-2 cells at low concentrations through passive diffusion, and they are not the substrates of P-glycoprotein. The transmembrane process did not consume energy and was not affected by temperature. The frying process reduced the absorption of P and PO; salt-soaking increased the absorption of P, IP and GPA; salt-frying enhanced the absorption of PO and GPA. Notably, adding salt facilitated the cellular uptake of the four components in Qing’e Pills. In summary, processing technology and auxiliary materials affected the absorption behavior of component compounds of the extract of Qing’e Pills, and will lead to correspondent change of blood concentration in vivo, eventually affect clinic efficacy.

Psoralen, isopsoralen and psoralenoside belong to coumarins. Psoralen can be transported across the MDCK-pHaMDR cell monolayer by passive diffusion and was moderately absorbed [25]. Meanwhile, most coumarins are all defined moderately intestinal absorbed compounds with Caco-2 cells [26,27,28]. This was consistent with our results, that it is more conducive to the absorption of compounds in neutral or weak alkaline small intestine environments. Compounds across cell monolayer are affected by molecular lipophilicity, hydrophilicity, molecular size and relative molecular weight. Also, the molecular structure directly determines the molecular polarity, the molecular size and the relative molecular weight to influence the absorption of compounds. The Caco-2 cell uptake mechanism of geniposidic acid, which belongs to iridoids, was passive diffusion [29], and there were no related reports about MDCK cells or hInEpCs [30]. 

Nevertheless, the reason why the herb processing method affected cellular uptake behavior of these 4 component compounds is still not clear. We speculate that this might be attributed to salt’s strong electrolyte character, which can affect the ionization of some components, result in changing of balance of ionic state and molecular state of these compounds, and further affect their cellular uptake behavior. This hypothesis needs to be investigated in future research. Based on the results, our next study objective will be to observe absorption characteristics of the four compounds between the in vivo and in vitro by other cells.

## 4. Materials and Methods 

### 4.1. Materials and Reagents

P, IP, PO, GPA, escoparone and rhein (internal standard, IS) with purity of 98% or higher were purchased from Nanjing World Biological Technology Co., Ltd. (Nanjing, China). UPLC-MS-grade methanol and acetonitrile were purchased from E. Merck (Merck, Darmstadt, Germany). HPLC-grade formic acid, and other analytically pure chemicals and solvents were obtained from Nanjing Chemical Reagent Co., Ltd. (Nanjing, China). Ultrapure water used throughout the experiments was prepared by Milli-Q ultrapure water purification system (Millipore Corporation, Bedford, MA, USA). Infrared thermometer TM750 (TECMAN, Nanjing, China). The 96-well plates (3599, corning-costar series, non-pyrogenic, tissue culture treated) and 24-well plates (3524, corning-costar series, non-pyrogenic, tissue culture treated) were purchased from corning Co. (New York, NY, USA). BCA Protein Assay Kit were purchased from Beyotime Biotechnology Co., Ltd. (Shanghai, China). MTT was purchased from Biosharp Inc. (BS030A, Nahui, China).

Raw *Cortex Eucommiae* was obtained from Nanjing Haichang Chinese Medicine Group Corporation (Nanjing, China), and raw *Fructus Psoraleae* was bought from Shanghai Leiyunshang Pharmaceutical Co., Ltd. (Shanghai, China). The traditional Chinese medicinal materials were identified as authentic by Jianwei Chen, College of Pharmacy, Nanjing University of traditional Chinese Medicine.

Caco-2 cells were purchased from ATCC (USA). Experiments were conducted on cells between passages 30–40.

### 4.2. Preparation of Extracts of Qing’e Pills 

The ratio of constituent herbs of Qing’e Pills was *Cortex Eucommiae*: *Fructus Psoraleae* 240 g, *walnut* 150 g and *garlic* = 4:2:1.25:1.

Processing of fried *Cortex Eucommiae* and *Fructus Psoraleae*: *Cortex Eucommiae* were stir-fried at 180 °C for 30 min till appearing scorched black color; *Fructus Psoraleae* were stir- fried at 150 °C for 15 min till deep brown color appeared (Chinese Pharmacopoeia, Vol. I, 2015). 

Processing of salt-soaked *Cortex Eucommiae* and *Fructus Psoraleae*: 100 g of *Cortex Eucommiae* was homogeneously wetted by 35 mL salt water (salt concentration was 6.7%) for 30 min; while 100 g of *Fructus Psoraleae* was homogeneously wetted by 20 mL of salt water (salt concentration was 10%) for 30 min, then air-dried in open.

Processing of salt-fried *Cortex Eucommiae* and *Fructus Psoraleae*: raw herbs of *Cortex Eucommiae* and *Fructus Psoraleae* were salt-soak processed as above. After that, *Cortex Eucommiae* was stir-fried at180 °C for 60 min, while *Fructus Psoraleae* was stir-fried at 150 °C for 15 min.

Processing of *walnut*: raw *walnut* was stir-fried at 80 °C for 10 min.

Processing of *garlic*: *garlic* was steamed over boiling water for 20 min.

All the above temperature was detected by using an infrared thermometer.

Preparation of Raw Qing’e Pills: raw constituent herb materials of *Cortex Eucommiae*, *Fructus Psoraleae*, *walnut* and *garlic* were mixed and crushed into fine powder. Then, the obtained powder was homogeneously mixed with honey at ratio of 2:1 to make pills (0.5 g per pill). Preparation of Fried Qing’e Pills: raw constituent herb materials of *Cortex Eucommiae*, *Fructus Psoraleae* and *walnut* were stir-fried as above described followed by powdering. Then, the obtained powder was homogenized with mashed *garlic* to make pills (0.5 g each).

Preparation of Salt-soaked Qing’e Pills: raw constituent herb materials of *Cortex Eucommiae* and *Fructus Psoraleae* were salt-soaked as above, then powdered and homogenized with mashed *walnut* and *garlic* to make pills same above. 

Preparation of Salt-fried Qing’e Pills: raw constituent herb materials of *Cortex Eucommiae* and *Fructus Psoraleae* were salt-fried as described above, followed by powdering. The obtained fine powder was homogenized with mashed *walnut* and *garlic* to make pills, same as above.

The extracts of the above variously prepared Qing’e Pills were obtained by following process. Firstly, to each of the Qing’e Pills sample, 50.0 g was weighed and decocted twice independently with 8-fold volume of water, 1 h each time. Secondly, the decoction was collected and filtrated through filter paper. Thirdly, 8-fold volume of 95% ethanol was added to the filtrate remains and reflux twice, 1 h each time. Finally, these two parts of filtrates were combined, condensed to a density of 1.0 g/mL by rotary evaporation at 55 °C and stored at −20 °C. Before using, proper amount of extract was diluted with medium, dissolved by ultra-sonication for 20 min, followed by centrifugation at 13,000 rpm for 5 min. The obtained supernatant was filtered with 0.22 μm filter membrane as the extract solution.

### 4.3. Cell Culture

Human intestinal cell line Caco-2 was routinely incubated in 75 cm^2^ cell culture flasks containing 10 mL of minimum essential medium (MEM) supplemented with 10% fetal bovine serum, 1% sodium pyruvate (100 mM), 1% nonessential amino acid and 1% penicillin-streptomycin (100 IU penicillin-100 g·mL^−1^ streptomycin) in a cell incubator at 37 °C with 95% air and 5% CO_2_. For routine maintenance, the medium was refreshed every 48 h and the cells were sub-cultured by 0.25% dissociation reagent when the confluency reached to 80–90%.

### 4.4. Standard Solution and Quality Control (QC) Samples

Stock solutions of P, IP, PO, GPA, escoparone and rhein (the latter two were internal standards) were prepared by dissolving appropriate weights of standards with methanol respectively. Working solutions for plotting calibration curves were prepared by mixing the stock solutions of the four components and serially diluted with methanol.

IS solution was obtained by diluting the stock solution with methanol to 1000 ng/mL for escoparone and 500 ng/mL for rhein. All the stock and working solutions were stored at 4 °C and returned to room temperature before using. 

Standard samples for plotting calibration curves were prepared by spiking 10 μL of IS and 10 μL of correspondent standard working solution into 40 μL of blank cell lysate within 50–2500 ng/mL for P, IP, PO and GPA. QC samples were prepared with the same process, to give three different concentration levels (100, 1000 and 2000 ng/mL) for P, IP, PO and GPA.

### 4.5. Biosample Preparation

Biosamples were prepared by spiking 10 μL of IS and 200 μL of methanol into 40 μL of cell lysate sample, followed by 5 min of vortexing. After centrifugation at 13,000 rpm for 5 min, the clear supernatant was transferred into 1.5 mL EP tube and evaporated to dry under nitrogen flow. The residue was re-dissolved by 50 μL of mobile phase and centrifuged again. Finally, 2 μL aliquot of the supernatant was injected into UHPLC-MS/MS system for analysis.

### 4.6. Apparatus and Operating Conditions

#### 4.6.1. Apparatus

The UHPLC–MS/MS system consisted of a triple quadrupole 5500 instrument (AB Sciex, Concord, Ontario, Canada) equipped with an electrospray ionization (ESI) source and a UHPLC system (Shimadzu, Tokyo, Japan) including an LC-30AD binary pump, an autosampler (Model SIL-30SD), an on-line degasser (DGU-20A5R) and a column temperature controller compartment (CTO-30A).

#### 4.6.2. UHPLC–MS/MS Conditions

Chromatographic separation was performed on an Extend C18 column (Agilent, USA, 2.1 mm × 100 mm, 1.8 μm) at 40 °C. The mobile phase consisted of 0.1% aqueous formic acid (A) and acetonitrile (B), which was eluted at a flow rate of 0.3 mL/min. The samples were stored at 4 °C before injection and the injection volume was set at 2 μL.

Positive-ion mode: The gradient program was as follows: 0.1–3.0 min, 15–45% B; 3.0–3.5 min, 45–50% B; 3.5–7.5 min, 50–65% B; 7.5–8.5 min, 65–85% B; 8.5–9.0 min, 85–15% B; 9.0–9.5 min, 15% B.

ESI-MS/MS analysis was operated in the positive ion mode under the following conditions: Curtain gas (CUR): 35 psi; ion spray voltage (ISV): 5500 V; temperature (TEM): 600 °C; ion source gas 1 (GAS1): 50 psi; ion source gas 2 (GAS2): 60 psi. The four components were quantified using the multiple reaction monitoring (MRM) mode with *m*/*z* 187.2/131.0 for P, IP, PO and *m*/*z* 207.2/151.1 for escoparone.

Negative-ion mode: The gradient program was as follows: 0.1–1.0 min, 10% B; 1.0–2.5 min, 10–85% B; 2.5–3.0 min, 85–90% B; 3.0–3.5 min, 90% B; 3.5–4.5 min, 10% B.

ESI-MS/MS analysis was operated in the negative ion mode under the following conditions: CUR: 35 psi; ISV: 5500 V; TEM: 550 °C; GAS1: 55 psi; GAS2: 55 psi. The two components were quantified using the MRM mode with *m*/*z* 373.0/211.3 for GPA and *m*/*z* 283.2/238.9 for rhein.

### 4.7. Validation of Developed UHPLC-MS/MS Method

#### 4.7.1. Specificity

The specificity of the method was studied by comparing the chromatograms of blank cell lysate, cell lysate sample spiked with analyte and IS, and real samples that were collected from cells, which were incubated with the drug that contained the medium of extracts.

#### 4.7.2. Linearity and Lower Limit of Quantification

Calibration curves were constructed by using spiked plasma samples at seven concentration levels. The linearity of each calibration curve was obtained by plotting the peak area ratio of each analyte to IS versus nominal concentration of analytes with a weighted (1/x^2^) least square linear regression. The lower limit of quantification was defined as the lowest concentration of calibration curve, which could be quantified reliably with a precision of below 20% and an accuracy of within ±20%.

#### 4.7.3. Precision and Accuracy

The accuracy and precision were detected by testing QC samples at low, middle and high concentrations respectively on three consecutive days. The intra-day and inter-day precisions were validated adequately. To study the precision, six duplicates were prepared. The accuracy should be within ±15% and the precision (RSD) should not exceed 15%.

#### 4.7.4. Extraction Recovery and Matrix Effect

The extraction recovery and matrix effect were assessed by analyzing QC samples at low, middle and high concentrations respectively. For each concentration, six duplicates were prepared. The extraction recovery was detected by comparing the peak areas of extracted QC samples with those of pure reference standards spiked in post-extracted blank rat plasma at the same concentration. Meanwhile, the matrix effect was evaluated using six duplicates by comparing the peak areas of extracted QC samples with those of samples in which blank plasma was replaced by water at equivalent concentrations. The RSD not surpassing 15% was considered satisfactory.

#### 4.7.5. Stability

The stabilities of three analytes were assessed under different conditions with QC samples at low, middle and high concentrations respectively. For each concentration, six duplicates were prepared. The conditions were as follows: three freeze-thaw cycles (−20 °C to room temperature), short-term stability (at room temperature for 4 h), and long-term stability (at −20 °C for 20 days). RSD should not exceed 15%.

### 4.8. Cell Viability Test

Caco-2 cells were seeded at 5 × 10^4^ cells cm^−2^ in a 96-well plate, cultured in an incubator for 24 h, and separated to control group and treat groups. After 24 h, the culture medium was removed. For the control group, 100 μL of MEM was added into each well; while for treat groups, 100 μL of drug-contained medium of various Qing’e Pills at different concentrations (5, 10, 20, 40, 80 mg/mL) was addedto each well. Five parallel wells were set for each concentration. Solutions from wells were removed after 48 h of incubation and 25 μL of MTT solution (5 mg/mL) were added. After 4 h of incubation, MTT solution was aspirated and 100 L DMSO were added for the solubilization of the formazan crystals. After solubilization, the absorbance was determined in a spectrophotometer at 595 nm [31,32,33]. The test was repeated three times. The inhibition rate was calculated by the following formula: Inhibition rate (%) = (OD_Control_ − OD_Drug_)/OD_Control_ × 100%(1)

### 4.9. Uptake by Caco-2 Cells

Caco-2 cells were seeded at 5 × 10^4^ cells cm^−2^ in a 24-well plate and cultured in an incubator for 14 d. During incubation, the culture medium was refreshed every other day. Firstly, the culture medium was discarded, 500 μL of HBSS was added to each well and incubated for 20 min at 37 °C. Secondly, each of the well was rinsed 3 times with HBSS at 37 °C and the fluid in well was pipetted to remove impurity on the cell surface. Thirdly, after adding 500 μL of drug-contained medium of extracts of various Qing’e Pills at different concentrations (three parallel wells for each concentration), the cells were incubated for indicated time. Fourthly, the drug-contained medium was discarded, and the cells were rinsed 3 times with HBSS at 4 °C to terminate cellular uptake. Fifthly, the cells were lysed by adding 150 μL of 0.5% Triton X-100 lysate and pipetted thoroughly. And finally, the cell suspension was transferred to a 1.5 mL EP tube and centrifuged at 13,000 rpm for 5 min to obtain the supernatant. Then 10 μL of the supernatant was used to measure the protein content by BCA kit, and the rest were determined for intracellular components by UHPLC-MS/MS.

Uptake = C/(Cproteins × h) (ug/g × h), C was the content of intracellular compounds (ng/mL), Cproteins was the content of intracellular protein (mg/mL), h was the time of uptake. The experimental results as cellular uptake (μg/g protein/h) were tested three times.

#### 4.9.1. Effects of Culture Media Containing Extracts with Different Concentrations on Cellular Uptake

Caco-2 cells were seeded at 5 × 10^4^ cells cm-2 in a 24-well plate. After 14 d of incubation, The effect of extract concentration on cellular uptake was evaluated by 60 min of incubation with 500 μL of culture media containing salt-fried Qing’e Wan with different concentrations (5, 40, 80, 120, 240 mg/mL).The treatment of sample was followed by procedures in “4.5”and “4.9”, content of components uptake were determined per unit protein and unit time.

#### 4.9.2. Effect of Temperature on Cellular Uptake

The results showed that the activity of most transporters was the highest at 37 °C and decreased gradually with the decrease of temperature. The activity disappeared at 4 °C [34]. In order to investigate whether the uptake behavior of the compounds in caco-2 cells depends on the transporter, the effect on uptake was investigated at 4 °C and 37 °C. One plate was placed in a cell incubator at 37 °C, with 5% CO_2_, and the other was in a 4 °C refrigerator. Caco-2 cells were seeded at 5 × 10^4^ cells cm^2^ in a 24-well plate. After 14 d of incubation, the effect of temperature on cellular uptake was investigated by 60 min of incubation with 500 μL of culture media containing salt-fried Qing’e Wan at 40 mg/mL. The treatment of sample was followed by procedures in “4.5”and “4.9”, content of components uptake was determined per unit protein and unit time.

#### 4.9.3. Effect of Inhibitor on Cellular Uptake

Caco-2 cells were seeded at 5 × 10^4^ cells cm^−2^ in a 24-well plate. After 14 d of incubation, the effect of inhibitor on cellular uptake was investigated by 60 min of incubation with 500 μL of culture media containing salt-fried Qing’e Wan (40 mg/mL) and inhibitors (concentrations of verapamil, cyclosporine A and sodium azide: 100, 10 and 500 μM respectively). The treatment of sample was followed by procedures in “4.5”and “4.9”, content of components uptake was determined per unit protein and unit time.

#### 4.9.4. Effect of pH on Cellular Uptake

Caco-2 cells were seeded at 5 × 10^4^ cells cm^−2^ in a 24-well plate. After 14 d of incubation, the effect of pH on cellular uptake was investigated by 60 min of incubation with 500 μL of culture media containing salt-fried Qing’e Wan at 40 mg/mL. Besides, the pH of culture medium containing salt-fried Qing’e Wan was adjusted to 5.0, 6.0, 7.0 and 8.0 by adding 1 M HCl or NaOH solution. The treatment of sample was followed by procedures in “4.5”and “4.9”, content of components uptake was determined per unit protein and unit time.

#### 4.9.5. Effect of Salt Concentration on Cellular Uptake

Caco-2 cells were seeded at 5 × 10^4^ cells cm^−2^ in a 24-well plate. After 14 d of incubation, the effect of salt concentration on cellular uptake was investigated by 60 min of incubation with 500 μL of culture media containing salt-fried Qing’e Wan with different salt concentrations (0.2, 0.5, 1.0, 2.0 and 5.0%) at 40 mg/mL. The treatment of sample was followed by procedures in “4.5”and “4.9”, content of components uptake was determined per unit protein and unit time.

#### 4.9.6. Effect of Herb Processing Methods on Cellular Uptake

Caco-2 cells were seeded at 5 × 10^4^ cells cm^−2^ in a 24-well plate. After 14 d of incubation, the effect of different processing methods on cellular uptake was investigated by 60 min of incubation with 500 μL of culture media containing raw, fried, salt-soaked and salt-fried Qing’e Wan at 40 mg/mL. The treatment of sample was followed by procedures in “4.5”and “4.9”, content of components uptake was determined per unit protein and unit time.

### 4.10. Statistical Analysis.

The results were expressed as the means ± standard deviation (SD). Statistical analyses were conducted by using one-way ANOVA of SPSS 21.0 software (SPSS Inc., Chicago, IL, USA). Multiple comparisons among all the groups were performed by using Dunnett multiple comparison test. Values of *p* < 0.05 were considered statistically significant.

## 5. Conclusions

The effects of drug-contained medium of different concentrations of extracts of Qing’e Pills on cells were investigated by using Caco-2 absorption cell model. Cellular uptake factors, including drug concentration, temperature, inhibitor, pH, salt concentration and processing methods (raw, fried, salt-soaked and salt-fried) were investigated in this study. After cell disruption and pretreatment, compounds that entered into the cells were detected by mass spectrometry. As a result, temperature and inhibitors showed no effect on cellular uptake for the 4 component compounds. In the range of 5–240 mg/mL, cellular uptake of the 4 component compounds increased with the increasing concentration of the extracts. In neutral and weakly alkaline environment, cellular uptake of the 4 component compounds increased along with the salt concentration, within the range of 0.2–2.0%. In summary, the salt-frying process of herb materials of Qing’e Pills, enhanced the cellular uptake of its main active component compounds.

## Figures and Tables

**Figure 1 molecules-23-03321-f001:**
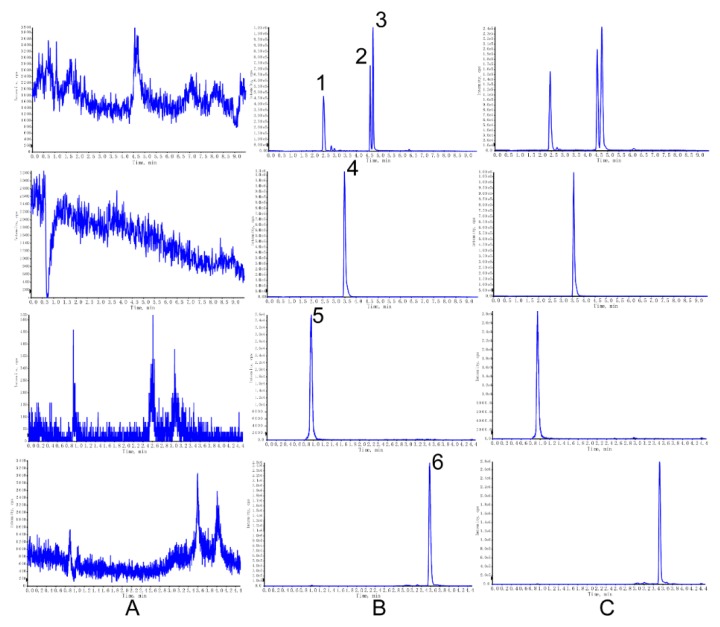
Representative multiple reaction monitoring (MRM) chromatograms of (1) Psoralenoside (PO), (2) Psoralen (P), (3) Isopsoralen (IP), (4) Escoparone, (5) Geniposide acid (GPA) and (6) Rhein in rat plasma. (**A**) Blank cell lysate, (**B**) cell lysate spiked with the standards and internal standard (IS), (**C**) sample collected from cells treated with standards.

**Figure 2 molecules-23-03321-f002:**
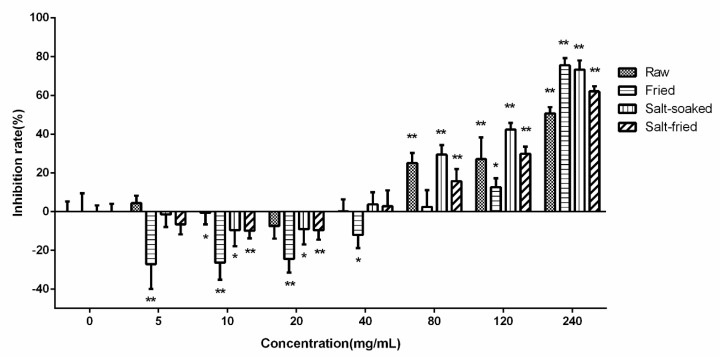
The effects of drug-contained medium with different concentration on Caco-2 cell proliferation activity. Significance: vs. blank medium: **p* < 0.05, ***p* < 0.01.

**Figure 3 molecules-23-03321-f003:**
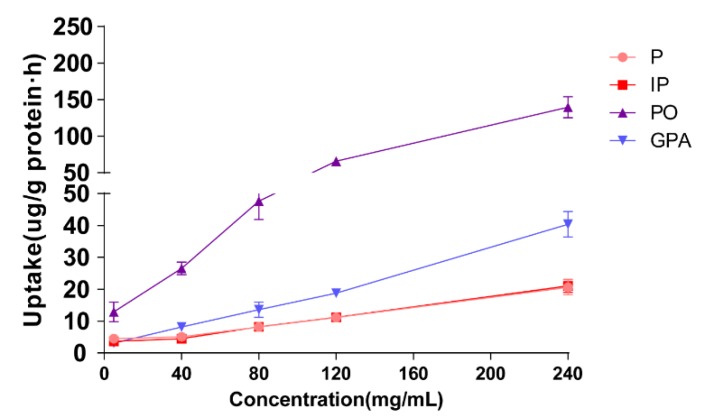
Cellular uptake of the 4 component compounds of various drug-contained medium of extracts of Qing’e Pills at different concentrations.

**Figure 4 molecules-23-03321-f004:**
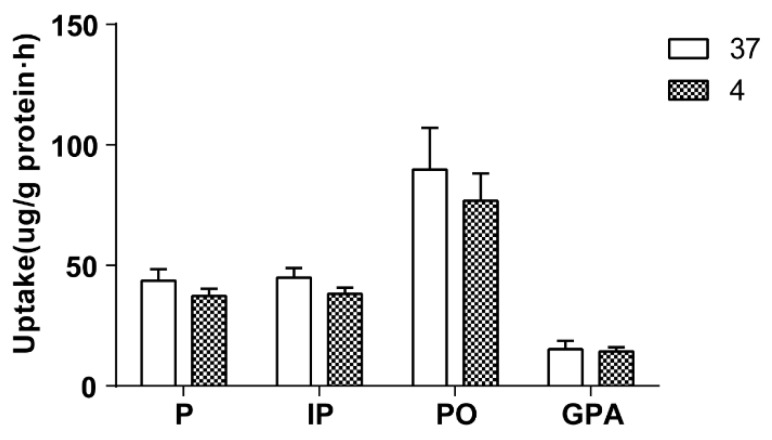
The uptake of the 4 component compounds at different temperatures.

**Figure 5 molecules-23-03321-f005:**
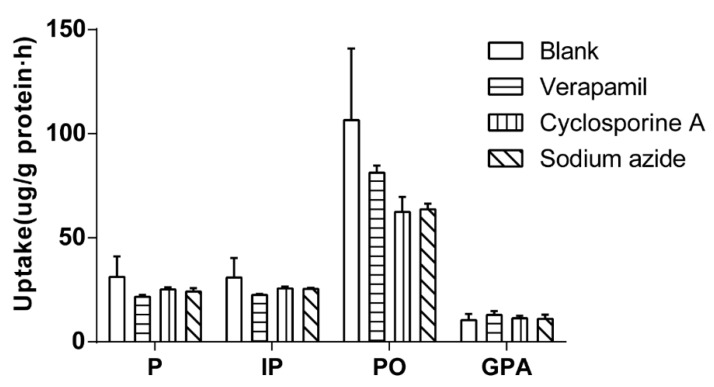
The cellular uptake of the 4 component compounds with the present of inhibitors.

**Figure 6 molecules-23-03321-f006:**
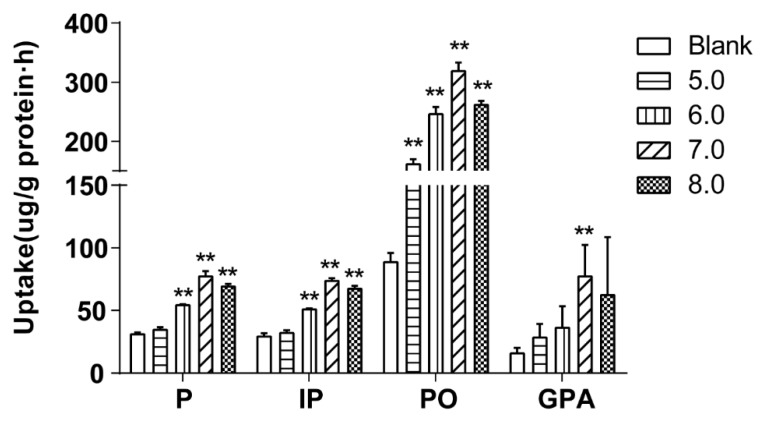
The cellular uptake of the 4 component compounds at different pH values. Significance: vs. blank medium: ** *p* < 0.01.

**Figure 7 molecules-23-03321-f007:**
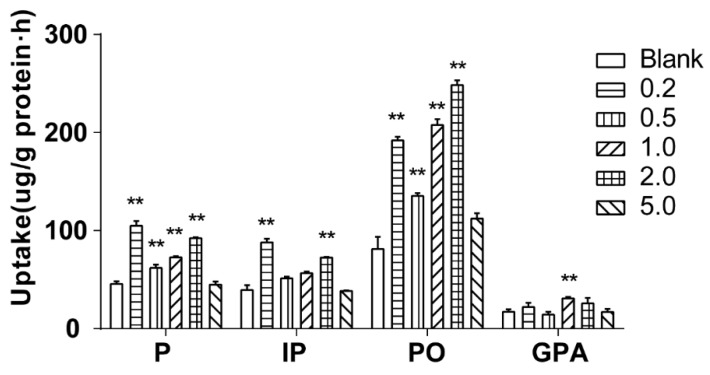
Cellular uptake of the 4 component compounds at different salt concentrations. Significance: vs. blank medium: ***p* < 0.01.

**Figure 8 molecules-23-03321-f008:**
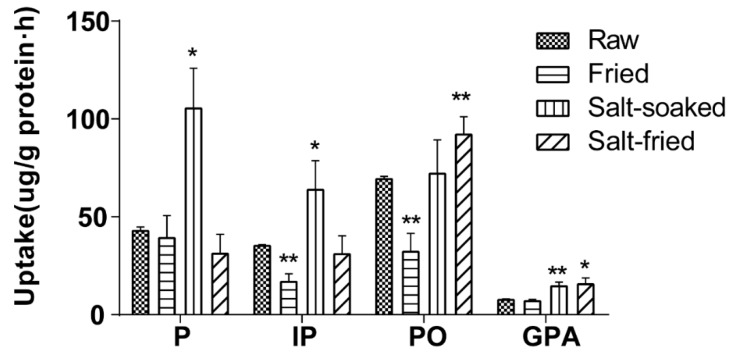
The cellular uptake of the 4 component compounds in drug-contained medium of various extracts of Qing’e Pills. Significance: vs. raw group (blank group): **p* < 0.05, ***p* < 0.01.

**Table 1 molecules-23-03321-t001:** Precision and accuracy of four analytes in cell lysate.

Analyte	Added Conc. (ng/mL)		Intra-Day			Inter-Day	
		Measured (ng/mL)	Precision (%)	Accuracy (%)	Measured (ng/mL)	Precision (%)	Accuracy (%)
P	10	9.65 ± 0.48	4.97	~3.47	9.71 ± 0.77	7.95	~2.87
	100	96.90 ± 5.42	5.59	~3.10	95.70 ± 5.29	5.53	~4.30
	400	390.92 ± 15.62	3.99	~2.27	382.98 ± 17.44	4.55	~4.26
IP	10	9.99 ± 0.77	7.72	~0.05	9.45 ± 0.91	9.60	~5.49
	100	96.44 ± 6.67	6.91	~3.56	97.83 ± 5.25	5.36	~2.17
	400	376.78 ± 13.06	3.47	~5.80	380.21 ± 22.16	5.95	~4.95
PO	10	9.35 ± 0.60	6.45	~6.54	9.36 ± 0.69	7.35	~6.39
	100	93.69 ± 6.86	7.32	~6.31	92.94 ± 6.86	7.39	~7.06
	400	378.12 ± 37.30	9.87	~5.47	380.85 ± 25.78	6.77	~4.79
GPA	10	9.48 ± 0.66	6.95	~5.23	9.52 ± 0.74	7.81	~4.78
	100	97.42 ± 7.21	7.40	~2.58	96.78 ± 8.14	8.41	~3.22
	400	380.40 ± 18.73	4.92	~4.90	379.85 ± 18.42	4.85	~5.04

**Table 2 molecules-23-03321-t002:** Recovery and matrix effect of analytes and IS.

Analyte	Added Conc. (ng/mL)	Extraction Recovery (%)	RSD (%)	Matrix Effect (%)	RSD (%)
P	10	91.85 ± 7.63	8.31	87.96 ± 3.85	4.37
	100	85.51 ± 5.95	6.96	92.24 ± 1.55	1.68
	400	89.07 ± 7.39	8.30	97.09 ± 6.95	7.16
IP	10	97.82 ± 4.98	5.09	92.42 ± 7.66	8.29
	100	77.19 ± 2.15	2.79	90.89 ± 2.66	2.92
	400	78.82 ± 4.75	6.03	85.40 ± 7.30	8.55
PO	10	80.92 ± 6.60	8.16	87.40 ± 7.63	8.73
	100	81.09 ± 3.89	4.79	84.82 ± 7.40	8.73
	400	87.26 ± 3.90	4.47	96.30 ± 5.84	6.06
GPA	10	82.11 ± 7.34	8.94	87.43 ± 5.30	6.06
	100	81.63 ± 5.13	6.28	84.01 ± 4.97	5.92
	400	79.58 ± 5.78	7.26	94.77 ± 3.71	3.91
Escoparone	10	88.02 ± 1.93	2.19	90.81 ± 4.50	4.96
	100	85.59 ± 3.12	3.64	92.32 ± 4.35	4.71
	400	88.69 ± 3.70	4.17	93.60 ± 3.91	4.18
Rhein	10	90.63 ± 3.88	4.29	92.54 ± 3.19	3.45
	100	95.16 ± 1.15	1.21	93.37 ± 3.24	3.47
	400	88.09 ± 3.90	4.43	92.86 ± 5.76	6.20

**Table 3 molecules-23-03321-t003:** Stability of four analytes in cell lysate.

Analyte	Concentration (ng/mL)	Three Freeze-Thaw Cycles		Short-Term Stability		Long-Term Stability	
		Measured (ng/mL)	RSD (%)	Measured (ng/mL)	RSD (%)	Measured (ng/mL)	RSD (%)
P	10	9.69 ± 0.53	5.45	9.78 ± 0.72	7.35	9.25 ± 1.01	10.9
	100	98.35 ± 1.45	1.47	96.17 ± 5.93	6.16	98.14 ± 7.09	7.22
	400	389.47 ± 42.05	10.8	385.09 ± 27.31	7.09	374.30 ± 24.02	6.42
IP	10	9.84 ± 0.26	2.69	9.81 ± 0.95	9.65	9.36 ± 0.80	8.57
	100	95.75 ± 8.97	9.37	95.76 ± 8.33	8.70	94.28 ± 9.05	9.60
	400	388.26 ± 6.30	1.62	383.88 ± 24.55	6.40	387.98 ± 25.72	6.63
PO	10	9.69 ± 0.94	9.72	9.58 ± 1.36	14.2	9.65 ± 1.14	11.9
	100	96.49 ± 7.73	8.01	94.61 ± 6.43	6.80	94.08 ± 9.27	9.85
	400	381.68 ± 18.74	4.91	385.48 ± 18.81	4.88	378.67 ± 35.33	9.33
GPA	10	9.09 ± 0.52	5.76	9.30 ± 0.73	7.81	9.63 ± 1.02	10.6
	100	95.25 ± 8.83	9.27	96.83 ± 10.48	10.8	95.38 ± 5.13	5.38
	400	367.97 ± 13.72	3.73	387.02 ± 14.78	3.82	384.28 ± 29.53	7.68

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
