# Peer review of "Evaluation of the Absorption Behavior of Main Component Compounds of Salt-Fried Herb Ingredients in Qing’e Pills by Using Caco-2 Cell Model"

_molecules, 2018, doi:10.3390/molecules23123321_

Round 1
Reviewer 1 Report
Submitted for review article ,,Evaluation of the Absorption Behavior of Main Component Compounds of Salt-fried Herb Ingredients in Qing’e Pills by Using Caco-2 Cell Model” is an original paper.
Generally, the topic of this article is interesting and concerns the impact of salt- frying on Caco-2 cell uptake of active ingredients in Qing'e Wan. The authors try to clarify whether this process affect the absorption of its main compounds. In this work the effects of culture media containing drugs of different concentrations of Qing'e Wan on cells were investigated by caco-2 cytotoxicity test. Although the human colorectal adenocarcinoma cell line is very often used in this type of research because of many advantages: are easy to culture and have the capacity to differentiate into cells with morphology and function of enterocytes or contain most drug-metabolizing enzymes, this model has some disadvantages. The main limitations are: the lack of mucus layer, playing the very important role in drug adsorption or lack the cellular heterogeneity. For this reason, the results obtained by authors should be compared with data from research conducted on alternative cell lines such as hInEpCs as a more physiologically relevant cell-based model. Abstract and Introduction are quite clear but in my opinion the materials and methods is too poor and unsatisfactory. In this work there are no methodological details such as protocols or references for a cytotoxicity test and others. The charts are illegible, especially representative MRM chromatograms (figure 1).The discussion seems to be too short. and poor not taking into account the available literature data about biologically active compounds of Qing'e wan and their effect on different cell lines. Why the authors did not investigate the effect of pure compounds (P, IP, PO, GPA, escoparone and rhein) on the Caco-2 cells?
In general, the work is interesting but the authors should correct it to publish it in Molecules
I have a few major comments which are submitted below:
- I suggest to conduct analogous studies on other cell lines to compare the results
- I suggest to improve the materials and methods section
- The authors should improve the charts
- Authors should polish up the English language and correct some tiny grammar mistakes.
- I suggest to examine the effect of pure compounds on the tested cell lines
Author Response
Response to Reviewer 1 Comments
Point 1: This model has some disadvantages. The main limitations are: the lack of mucus layer, playing the very important role in drug adsorption or lack the cellular heterogeneity. For this reason, the results obtained by authors should be compared with data from research conducted on alternative cell lines such as hInEpCs as a more physiologically relevant cell-based model.
Response 1: It’s a very good advice. We will use other cell lines like hInEpCs ,MDCK cell for comparison in future research. Caco-2 cell is a generally recognized in vitro cell model for absorption research. Besides, in our other work, we have used the extracts of Qing’e Pills perfused rat small intestine (duodenum, jejunum, ileum and colon) in vivo and investigated
the difference of absorption in these four intestinal segments and its effect on the absorption of active components in Qing'e pills.
Point 2: The materials and methods is too poor and unsatisfactory. In this work there are no methodological details such as protocols or references for a cytotoxicity test and others.
Response 2: The content of the Chapter of “materials and methods” have been enriched by providing with detailed protocols.
Point 3: The charts are illegible, especially representative MRM chromatograms (figure 1).
Response 3: The horizontal and vertical coordinates of figure 1 have been enlarged.
Point 4: The discussion seems to be too short. and poor not taking into account the available literature data about biologically active compounds of Qing'e wan and their effect on different cell lines.
Response 4: The details have been supplemented in the article.
Point 5: Why the authors did not investigate the effect of pure compounds (P, IP, PO, GPA, escoparone and rhein) on the Caco-2 cells?
Response 5: The objective of this research was trying to find out whether herb processing method affects the absorption behavior of their component compounds, so that to provide useful information for TCM clinic or pharmaceutical. Pure compound cannot provide this kind of information.
Point 6: Authors should polish up the English language and correct some tiny grammar mistakes.
Response 6: The English language has been polished up in this revised manuscript.
The above changes will not influence the content or framework of the paper. We have marked the changes in the revised manuscript. Besides, considering of consistency with the Tittle, we have changed “Qing’e Wan” (“Wan” is the Chinese pronunciation of “Pills”) to “Qing’e Pills” in the revised manuscript.
We highly appreciate for Editors/Reviewers’ work, and hope that the corrections will meet the requirements of possible acceptance.
Once again, thank you very much for your comments and suggestions.
Thank gain for all of your comments and good suggestions.

Reviewer 2 Report
Authors introduced the oriental medicine absoption depending on several factors. This manuscript is out of scope in Molecules. I would like to suggest to select the analytic journals or chemistry related journals.
Author Response
Response to Reviewer 2 Comments
Point 1: Authors should polish up the English language and correct some tiny grammar mistakes.
Response 1: Both of the grammar and wordings of English language in this article have been polished up.
The above changes will not influence the content or framework of the paper. We have marked the changes in the revised manuscript. Besides, considering of consistency with the Tittle, we have changed “Qing’e Wan” (“Wan” is the Chinese pronunciation of “Pills”) to “Qing’e Pills” in the revised manuscript.
We highly appreciate for Editors/Reviewers’ work, and hope that the corrections will meet the requirements of possible acceptance.
Once again, thank you very much for your comments and suggestions.
Thank gain for all of your comments and good suggestions.
Reviewer 3 Report
The manuscript describes an investigation on the effects of different variables on the uptake of compounds from Qing'e Wan traditional Chinese medicine into Caco-2 cells. The following revisions are needed:
1) The abstract does not capture the key aspects of the study (e.g. no mentioning of variables such as salt concentration, temperature, pH etc are made in the abstract). It should be elaborated to incorporate all the variables that were investigated in order to give the reader a glimpse of the important work that was done in this study.
2) The methods are incomplete. The following details should be given in the methods:
2.1) Page 10, line 222: Explain in detail what is meant by "fried" according to the Chinese Pharmacopoeia. Please give the conditions under which the plant materials were fried including temperature and time of exposure and any other procedures that were followed during this process.
2.2) Page 10, lines 220 and 229: What "flavors" were made into pills? Also give the method on how the pills were prepared.
2.3) Page 10, line 244: Does "fuse" mean "confluency"?
2.4) Page 11, lines 329: Please give details on the plates that were used, e.g. were they tissue culture treated?
2.5) Page 11, line 330: Explain why the cells were only grown for 14 days? Caco-2 cells are ussually grown for 21 days to reach confluent monolayers. How did you determine monolayer formation?
2.6) Page 12, line 341: How was the protein content of the cells determined in oprder to express the uptake as "ug/g protein/h"?
2.7) Please elaborate on the cell uptake studies to describe how the variables were investigated including effect of temperature, salt concentration, pH etc. It is very important to provide sufficient information to the reader to understand how the experiments were conducted, e.g. the temperature of what was changed? This is also applicable to all the other variables.
3) Page 5, lines 103-109: Cytotoxicity is interchangeably used with proliferation and inhibition in this section. Please describe clearly what was determined in this experiment (e.g. cell viability) and how this relates to cytotoxicity. Also rather use extracts and isolated compounds instead of "drugs".
4) Page 6, line 114 and onwards: Most of the headings in section 2.4 are not at their correct places, e.g. 2.4.1 states "Effect of temperature on cellular uptake" but the "effect of drug concentration" is discussed in the paragraph below.
5) Page 9, line 181: The sentence does not make sense and must be re-written.
6) Page 9, line 190: Please elaborate on how the conclusion could be made that no energy was consumed. If this is only based on linear absorption curves based on concentration increases, then it should be noted how this was distinguished from facilitated uptake that will also be linear until the carriers are saturated.
7) English grammar and some sentences need to be revised in terms of language use.
Author Response
Response to Reviewer 3 Comments
Point 1: The abstract does not capture the key aspects of the study (e.g. no mentioning of variables such as salt concentration, temperature, pH etc are made in the abstract). It should be elaborated to incorporate all the variables that were investigated in order to give the reader a glimpse of the important work that was done in this study.
Response 1: The abstract of the article has been refined.
Point 2: The methods are incomplete. The following details should be given in the methods:
2.1) Page 10, line 222: Explain in detail what is meant by "fried" according to the Chinese Pharmacopoeia. Please give the conditions under which the plant materials were fried including temperature and time of exposure and any other procedures that were followed during this process.
Response 2.1: Here, fried means “stir fried”. The detailed protocols were described in revised manuscript.
Point 2.2: Page 10, lines 220 and 229: What "flavors" were made into pills? Also give the method on how the pills were prepared.
Response 2.2: “Flavors” is a misused word for herb mixture. The wording has been corrected. Besides, the protocol of Qing’e Pills preparation was described in revised manuscript.
Point 2.3: Page 10, line 244: Does "fuse" mean "confluency"?
Response 2.3: Yes, the wording has been corrected.
Point 2.4: Page 11, lines 329: Please give details on the plates that were used, e.g. were they tissue culture treated?
Response 2.4: Yes, they were tissue culture treated. The 96-well plates (product number: 3599) and 24-well plates (product number: 3524) used in experiment were produced by USA Corning Company. They were all in the corning-costar series. They were made of polystyrene and non-pyrogenic.
Point 2.5: Page 11, line 330: Explain why the cells were only grown for 14 days? Caco-2 cells are ussually grown for 21 days to reach confluent monolayers. How did you determine monolayer formation?
Response 2.5: In this experiment, Caco-2 cells were cultured for uptake experiment, and monolayer cells were formed after 14 days of culture by verification. Monolayer cells that have been formed was determined by electron microscope (×200). In this experiment, the plates we used were not transwell plates and could not measure the membrane potential of monolayer cells.
Normally, the TEER value was used to determine the monolayer formation of Caco-2 cells on a transwell membrane. In this research, we did not aim to form an intact Caco-2 cell monolayer. We just detected the amount of compounds that entered into Caco-2 cells themselves. For this reason, with 14 days of cell culture, Caco-2 cells reached 80%~90% confluency. Results manifested, this method served our purpose right well.
Point 2.6: Page 12, line 341: How was the protein content of the cells determined in order to express the uptake as "ug/g protein/h"?
Response 2.6: The unit “ug/g protein/h” represents the amount of component compound that absorbed by per gram of cells (represented by the weight of protein) within 1 h. The BCA protein assay is a popular colorimetric detection method for quantification of protein. The amount of cells can be represented by the amount of protein. Unit time is calculated because content of compound in cell uptake was related to time of uptake.
Point 2.7: Please elaborate on the cell uptake studies to describe how the variables were investigated including effect of temperature, salt concentration, pH etc. It is very important to provide sufficient information to the reader to understand how the experiments were conducted, e.g. the temperature of what was changed? This is also applicable to all the other variables.
Response 2.7: The effect on cell uptake for the mentioned variables was described in revised manuscript. The results showed that the activity of most transporters was the highest at 37 °C and decreased gradually with the decrease of temperature. The activity disappeared at 4 °C. In order to investigate whether the uptake behavior of the compounds in caco-2 cells depends on the transporter, the effect on uptake was investigated at 4 °C and 37 °C.
Point 3: Page 5, lines 103-109: Cytotoxicity is interchangeably used with proliferation and inhibition in this section. Please describe clearly what was determined in this experiment (e.g. cell viability) and how this relates to cytotoxicity. Also rather use extracts and isolated compounds instead of "drugs".
Response 3: ① Cytotoxicity primarily indicates the inhibitory effect of extracts on cell viability. In the formula, the inhibition rate was expressed as: Inhibition rate = (ODControl-ODDrug)/ODControl ×100%. ② The wordings were reconsidered. Drugs were replaced by extracts in the revised manuscript.
Point 4: Page 6, line 114 and onwards: Most of the headings in section 2.4 are not at their correct places, e.g. 2.4.1 states "Effect of temperature on cellular uptake" but the "effect of drug concentration" is discussed in the paragraph below.
Response 4: Those errors have been corrected in the revised manuscript.
Point 5: Page 9, line 181: The sentence does not make sense and must be re-written.
Response 5: The sentence has been re-written in the revised manuscript. The structural and biochemical functions of Caco-2 cell (the human colon adenocarcinoma cell lines) are similar to that of human intestinal epithelial cell, with enzyme system similar to those of the brush-edge epithelium of the small intestine.
Point 6: Page 9, line 190: Please elaborate on how the conclusion could be made that no energy was consumed. If this is only based on linear absorption curves based on concentration increases, then it should be noted how this was distinguished from facilitated uptake that will also be linear until the carriers are saturated.
Response 6: Within the range of extract concentrations involved in the experiment, the cellular uptake of tested component compounds did not affected by energy inhibitors and P-glycoprotein inhibitors. Based on this result, we made the conclusion.
Point 7: English grammar and some sentences need to be revised in terms of language use.
Response 7: English grammar and some sentences have been corrected in the revised manuscript.
The above changes will not influence the content or framework of the paper. We have marked the changes in the revised manuscript. Besides, considering of consistency with the Tittle, we have changed “Qing’e Wan” (“Wan” is the Chinese pronunciation of “Pills”) to “Qing’e Pills” in the revised manuscript.
We highly appreciate for Editors/Reviewers’ work, and hope that the corrections will meet the requirements of possible acceptance.
Once again, thank you very much for your comments and suggestions.
Thank gain for all of your comments and good suggestions.

Round 2
Reviewer 1 Report
I accept changes made by authors.
Reviewer 2 Report
This paper presented some molecules from herb medicine and the re-treatment.
We can't find out any new information. If there are, authors should emphasize the novelity and effects on the readers. In addition, any interesting point could not be found. I would suggest the authors introduce the significance of retreatment of herb ingredients, the new points authors found from this experiment (which I feel, is not new), and the perspective from the results.
Reviewer 3 Report
The reviewers' comments have been addressed satosfactory.